# The Main Failure Modes of Hot-Work Die Steel and the Development Status of Traditional Strengthening Methods and Nano-Strengthening Technology

**DOI:** 10.3390/ma17143455

**Published:** 2024-07-12

**Authors:** Hong-Yu Cui, Ze-Ju Bao, Qin Gong, Shi-Zhe Bao, Yun-Zhi Zou, Ai-Min Li, Hong-Yu Yang, Cheng-Gang Wang, Zhi-Gang Li, Fang Chang, Shi-Li Shu, Jie Kang, Ming Zhu, Feng Qiu, Qi-Chuan Jiang

**Affiliations:** 1Key Laboratory of Automobile Materials, Ministry of Education and Department of Materials Science and Engineering, Jilin University, Renmin Street No. 5988, Changchun 130025, China; cuihy1621@mails.jlu.edu.cn (H.-Y.C.); gongqin1621@mails.jlu.edu.cn (Q.G.); baosz1621@mails.jlu.edu.cn (S.-Z.B.); zouyz1621@mails.jlu.edu.cn (Y.-Z.Z.); yanghongyu2021@jlu.edu.cn (H.-Y.Y.); changfang@jlu.edu.cn (F.C.); jiangqc@jlu.edu.cn (Q.-C.J.); 2Zhanghang Shangda Superalloys Materials Co., Ltd., No. 16 Wangong Avenue, Qinghe County, Xingtai 054800, China; 3FAW Foundry Co., Ltd., Changchun 130013, China; cgwang0611@163.com (C.-G.W.);; 4School of Mechanical and Aerospace Engineering, Jilin University, Renmin Street No. 5988, Changchun 130025, China; shushili@jlu.edu.cn; 5Jilin Liyuan Precision Manufacturing Co., Ltd., No. 5729, Xi’ning Road, Economic Development Zone, Liaoyuan 136299, China; xingzheng@liyuanjingzhi.net.cn; 6Zhenjiang Xianfeng Automotive Parts Co., Ltd., Dantu High Tech Industrial Park, Zhenjiang 212000, China; ming.zhu@xfelectronic.cn

**Keywords:** hot-work die steel, failure modes, traditional strengthening methods, nano-strengthening technology

## Abstract

As an important part of die steels, hot-work die steels are mainly used to manufacture molds made of solid metal or high-temperature liquid metal from heating to recrystallization temperature. In view of the requirements for mechanical properties and service life for hot-work die steel, it is conducive to improve the thermal fatigue resistance, wear resistance, and oxidation resistance of hot work die steel. In this review, the main failure modes of hot-work die steel were analyzed. Four traditional methods of strengthening and toughening die steel were summarized, including optimizing alloying elements, electroslag remelting, increasing the forging ratio, and heat treatment process enhancement. A new nano-strengthening method was introduced that aimed to refine the microstructure of hot-work abrasive steel and improve its service performance by adding nanoparticles into molten steel to achieve uniform dispersion. This review provides an overview to improve the service performance and service life of hot work die steel.

## 1. Introduction

Hot-work die steels are suitable for manufacturing dies for the hot deformation of metals, such as hot extrusion dies, hot-forging dies, hot upsetting dies, and die-casting dies [1]. With the urgent demand for lightweight in the automobile, high-speed train, aerospace, and other industries and the rapid development of advanced intelligent manufacturing, higher requirements have been put forward for the mechanical properties of hot-work die steel. High-quality hot-work molds under severe conditions such as thermal erosion of high-temperature molten metal and hot and cold cycles are mainly die-casting molds, low-pressure casting molds, gravity casting molds, hot extrusion molds, forging molds, and hot injection molds in pillar industries such as automobiles, machinery, and electronics. The current failure modes are divided into the following categories: thermal fatigue cracking, thermal erosion, and wear. In addition, thermal fatigue resistance and wear resistance are the key factors affecting the life of hot-work molds, and have always been challenging problems in the international hot-work mold field [2,3,4].

Since the 1920s, efforts have been made to optimize composition design of hot-work die steel. After four development stages, when the contents of Cr and Mo (wt.%) were increased from 1 to 5 and 0.2 to 2.5, respectively, the mechanical properties of hot-work die steel were improved, but no significant breakthrough was achieved. The main reason is that the traditional strengthening methods such as metal liquid purification, composition design, micro-alloying, incubation and deterioration, heat treatment, and large plastic deformation [4] no longer meet the requirements of engineering applications of metal materials. Therefore, in order to develop high-performance metal materials, an innovative path to strengthen hot-work die steel technology is urgently needed.

## 2. Study on Main Service Failure Forms and Mechanisms of Die Steel

The precision mold parts are typically made from high-quality materials that possess excellent ductility, toughness, fatigue strength, wear resistance, and corrosion resistance, as well as hardness, machined type, polishing property, and dimension stability based on the intended application [5]. Since it is difficult to appropriately select the maximum abrasion resistance, best resistance to high temperature softening, and best toughness, all the factors should be determined prior to choosing a material [6]. Typically, the changes in shape, size, and material characteristics can reduce the property of die, leading to the occurrence of a failure. The failure mechanism of mold steel is related to the working conditions [7]. In addition to the normal mechanical load, the hot mold is also subjected to repeated heating and cooling cycles, making the material more susceptible to heat erosion, heat crack, heat fatigue, heat wear, and plastic deformation [8].

### 2.1. Study on Fatigue Behavior of Die Steel

Hot=work die steel, as the basic material of the mold industry, is widely used in high-pressure die casting, hot extrusion, hot forging, hot stamping, hot rolling, milling, hot piercing, and other fields. The hot-work mold cavity can withstand high pressure and temperature that even exceeds 873 K, and the thermal cycle will repeat thousands of times. Most the effects of thermal fatigue damage on the failure mechanisms are manifested as the loss of temper stability and high-temperature fatigue strength. The heating and cooling cycle process on the surface of the mold steel will directly produce a temperature gradient in the mold, which causes a compressed state during the heating process and in a stretched state during the cooling process for the mold. The alternating stretching–compression state will cause a deformation in the mold. The cyclic stress will eventually cause cracks on the mold surface with an increasing number of cycles due to the production of elastic and plastic strains caused by thermal stress [9,10,11,12,13].

Li et al. [9] investigated the initiation and propagation mechanism of thermal fatigue cracks in AISI H13 (ISO 4957: X40CrMoV5-1 [10]) steel based on the Uddeholm thermal fatigue test and the micromorphology of thermal fatigue cracks on the surface of H13 steel specimens with an initial hardness of 46HRC after different times of thermal cycles. Figure 1a is a schematic diagram of the sample used in the experiment and Figure 1b shows the change in surface hardness after multiple heating and cooling cycles. It can be seen that tempering transformation and obvious softening occur, which should be closely related to the number of thermal cycles. As the number of thermal cycles increases, the surface hardness of the sample decreases significantly and the softening depth increases. Figure 1c–e shows the optical microscope (Figure 1(c_1_–e_1_)), SEM (Figure 1(c_2_–e_2_)), and TEM (Figure 1(c_3_–e_3_)) microstructure images of H13 steel under different cycle times. After 100 thermal cycles, slight wrinkles appeared on the surface of the specimen. After 1500 thermal cycles, thermal fatigue cracks began to expand both transversely and longitudinally along the specimen, intertwining with each other to form network cracks. After 3000 thermal cycles, the thermal fatigue cracks of H13 steel further expanded under thermal load conditions. The main cracks have penetrated the entire observation area, and the fine carbide precipitates have almost disappeared. As the number of thermal cycles increases, significant coarsening of the carbides can be observed. In addition, most of the precipitates are irregular spherical and rod-shaped carbides, which are mainly M_23_C_6_ and M_6_C type carbides.

Klobčar et al. [11] developed an immersion testing machine for simulating thermal fatigue testing under actual service conditions for the aluminum alloy die-casting process. Figure 1f shows the SEM images of thermal fatigue cracks in the test steel after 20,000 cycles of immersion testing and Figure 1(f_1_–f_5_) shows EDS images of chemical elements around thermal fatigue cracks. As can be seen, there are higher Al and Si concentrations at the crack tip, indicating that the crack tip is filled with molten aluminum alloy. Additionally, the traces of Cr and O are also found at the crack tip. Furthermore, the occurrence of higher concentrations of Cr in the oxide layer around the cracks shows that the crack tip is oxidized due to the presence of oxygen and the high-temperature environment. Thermal crack formation occurs though nucleation, initial growth, and continued crack propagation, which results in surface damage or surface delamination of the decomposed material. Crack nucleation is not only associated with the accumulation of local plastic strains in the surface material but also the welding of aluminum alloy to the tool surface followed by material washout. This, in turn, creates sharp crack nucleation in the surface. The increase in the volume of the oxide layer, that is, the oxidation of the crack surface, promotes the initial growth of thermal fatigue cracks. The crack-filling in the casting material as well as the oxidation and softening of the mold material due to tempering or aging of the surface layer all contribute to accelerated crack growth [14].

Figure 1g–i shows the mechanism of oxidation-assisted growth of hot crack. The iron atoms in the steel diffuse to the surface and contact with oxygen. The vacancies of iron atoms are filled with alloying elements, and then a layer of iron oxide usually forms on the surface of the mold steel and cracks. In addition, aluminum and silicon oxides are also present. On the one hand, the tensile stress generated by the atomic migration in the oxide layer during hot and cold cycles can lead to the initiation of cracks. On the other hand, the low thermal expansion coefficient, large volume, and brittleness of the oxide layer, and especially the presence of nitrogen and aluminum alloy inside the cracks of the mold during operation, can also lead to the initiation of cracks [13,15,16]. In summary, the increase in the tension at the crack tip during cyclic cooling causes the growth of the crack, and the oxidation of the crack surface of die steel promotes the expansion of cracks. It is reported that the oxidation behavior of the die steel seriously affects its service life [17].

### 2.2. Study on Wear Behavior of Die Steel

Hot-work die steels are mainly used to manufacture dies for the processing of materials at high temperature and high pressure, including hot forging, hot extrusion, and die-casting [18]. The serious friction and wear can occur between the hot-work die steel and the hot deformation material at high temperatures, which leads to the changes in the geometric dimensions of the workpiece and a weakness in surface quality. Moreover, there is friction between the blank and mold, which causes the uneven deformation and damage to the mold. Improving the high-temperature friction and wear resistance of hot-work die steel can extend the service life of the die and improve the quality of the workpiece, which is of very important research significance.

The high plastic deformation resistance and processing temperatures during service have a particularly serious effect on the wear and failure behavior of hot extrusion molds. For example, Wang et al. [19] studied the wear of the H13 die surface during the extrusion process. Figure 2a,b show the microscopic images of longitudinal sections of die steel after hot extrusion. It can be seen that a large number of cracks appear in both the surface oxide layer and the internal oxide layer. Some of the surface oxide layers are peeled off to form pits, and the granular microstructure of the internal oxide layer provides a platform for obtaining a large number of holes. The oxygen atoms diffuse from the saturated surface to the interior caused by extrusion at high temperatures, causing oxidative wear on the surface and interior of the mold. Figure 2c shows a schematic diagram of the mechanism of oxidation wear on the surface of die steel. The formation of deep cracks and pits indicates that the grain boundaries in the oxide layer are severely oxidized. In the matrix, microcracks initiate at the grain boundaries, motivating the occurrence of the oxidation reaction. The internal oxide layer transformed from the matrix promotes the formation of holes and cracks at the grain boundaries as well as the subsequent expansion of cracks. Conversely, the initiation and expansion of cracks also promote the oxidation of the crack surface [19,20].

There are several factors that affect the friction and wear resistance of die steel at high temperatures, including temperature, load, and rotation speed, etc. Zhou et al. [20] studied the wear behavior of tungsten hot-work die steel (AISI H21) under different experimental conditions. Figure 2d–i show the cross-sectional morphology of H21 steel at different wear conditions. They found that die steel has a good wear resistance at 400 °C, while its wear resistance deteriorates as there is an increase in temperature. In particular, the serious oxidation wear appears when the temperature increases from the room temperature to 600 °C. The frictional oxidation inevitably occurs when metal alloys slide at high temperatures, which leads to the formation of a friction oxide layer on the wear surface. The tribo-oxide layer is considered to be a decisive factor in determining the wear behavior and even the wear mechanism.

The wear resistance of hot-work die steel is closely related to the oxidation resistance that is closely related to the Cr content. Cheng et al. [21] studied the effect of a prefabricated Cr_2_O_3_ layer on the wear resistance of steel and found that a layer of Cr_2_O_3_ is first formed on the surface through high-temperature oxidation (as shown in Figure 2j,k). The grinding disc without high-temperature oxidation initially has adhesive wear, which will decrease accordingly as the wear progresses, due to the formation of Cr_2_O_3_ particles. This Cr_2_O_3_ layer can not only reduce the direct contact between the metal pin and the grinding disc and adhesive wear, but also change the wear and friction behavior. Therefore, this Cr-rich oxide layer plays a very active and effective role in improving anti-friction and wear at high temperatures.

However, the wear resistance of mold steel cannot only be judged by its oxidation resistance, and the content of the element is also considered. Li et al. [22] compared the friction and wear characteristics of new Mo-W hot-work die steel (SDCM-S) and H13 steel at high temperatures. They found that the oxidation resistance of steel increases with the increase in Cr content. Therefore, the oxidation resistance of H13 steel is much higher than that of SDCM-S steel because the Cr content of H13 steel is higher than that of SDCM-S steel and the formation of a Cr_2_O_3_ oxide layer on the surface. However, SDCM-S steel exhibits better tempering stability, which delays the transition from light wear to heavy wear and can maintain light oxidation wear at high temperatures for a long time. Therefore, the wear rates of SDCM-S at 400 °C and 700 °C are lower than that of H13 steel (as shown in Figure 2l). In addition, the wear rate of H13 increases more obviously with increasing temperature.

### 2.3. Study on Oxidation Behavior of Die Steel

The service time of molds at high temperatures strongly depends on their oxidation resistance. Therefore, improving the oxidation resistance of the die steel will significantly improve the service life of die steel [23,24].

It is reported that the oxidation mechanism of steel is parabolic oxidation. The resulting oxide skin is made up of the following three layers: the outer layer, the transition layer, and the inner layer, The composition of the oxide layer is related to the alloy elements actually present in steel, but Cr_2_O_3_, SiO_2,_ and Al_2_O_3_ are generally considered as protective oxides [25,26,27]. The long-term oxidation resistance of die steel is mainly determined by the compactness of oxide layer, good adhesion, and slow growth rate. The elements with high content in the alloy are basically oxidized during the high temperature oxidation process, and the amounts of various oxides in the oxide layer are approximately proportional to the concentration of the elements in the alloy. The element Cr gives the steel high oxidation resistant, which is mainly due to the Cr_2_O_3_ produced during oxidation, which provides a barrier against further oxidation. The chromium oxide is divided into two layers, Cr_2_O_3_ and MCr_2_O_3._ The compacted Cr_2_O_3_ layer adheres well to the substrate. The MCr_2_O_3_ layer, namely, a spinel type oxide, is brittle and weak to the inner layer and, therefore, prone to shedding. Some studies have shown that the effect of grain size on the high-temperature oxidation resistance of alloys is related to the initial Cr content of Cr-containing steel. For low-Cr steel (less than 2.25 wt.%), an increase in the grain size can improve antioxidant resistance, while steels with high Cr content are more likely to form thin protective Cr_2_O_3_ on small grain surfaces. 

The formation of an oxide layer is a chemical reaction process, and the thickening of the oxide layer is caused by a combination of diffusion and chemical reactions. Because of the large radius and high mobility of oxygen ions, the oxidation layer can often grow epitaxially. The porous oxide layer caused by internal stress and volatile oxide volatilization is the focal point of crack initiation stress [25]. Zhang et al. [23] compared the oxidation behavior of 3Cr3Mo2NiW and 3CrNi3Mo steel in air at 600 °C. They found that the internal oxide layer of 3Cr3Mo2NiW steel formed a dense chromium-rich oxide layer after 10 h of oxidation, which effectively prevented the continuation of oxidation. The internal oxide layer in 3CrNi3Mo steel formed an adhesion layer with a reticulated structure composed mainly of Ni- and Cr-rich spinel oxides, without forming a barrier against oxidation (as shown in Figure 3a,b).

Salem et al. [28] studied the effects of plating on aluminum and oxidation on thermal fatigue damage of high-pressure die-casting mold steel. They found that a thick oxide layer was formed on the surface of the original steel after 30,000 cycles of thermal fatigue, consisting of an internal layer rich in chromium oxide and an external layer poor in chromium oxide. Figure 3c,d show that microcracks propagate from surface thermal cracking into the matrix through the oxide scale and all the microcracks are filled with internal chromium oxide. Microcracks also appeared in the surface oxidation layer. When the steel is coated with aluminum, the chemical elements propagate along the surface of the coating and form Al_2_O_3_, Fe_2_O_3_, Fe_3_O_4_, and (Fe, Cr)_3_O_4_, while the crack tip mainly forms a Cr_2_O_3_ oxide layer, as shown in Figure 3e. Finally, it is concluded that the crack initiation mechanism is mainly determined by oxidation and surface layer characteristics. After hundreds of thousands of thermal cycles in the air, microcrack networks are formed on the surface of the original steel and coated steel. The short-circuit diffusion of oxidation through the substrate steel leads to the propagation of hot crack networks in the steel. Oxidation also promotes the diffusion of microcracks to the intermetallic compound layer. In addition, nitrogen action can significantly reduce the oxidation of raw steel and coating samples and delay microcrack initiation. This shows that microthermal cracking plays a decisive role in the early cracking of steel, while high-temperature oxidation contributes to the crack propagation. 

Surface morphology plays an important role in the oxidation behavior of hot-work die steel. Because of that, parameters such as amplitude, ordering, or directionality provide significant information on the surface condition of a material used in the power industry. Numerical methods, mainly the fractal analysis, are important tools to characterize structures of oxide layers and their properties after long-term operation at elevated temperatures. The studies carried out on three different steel grades used in the power industry (13CrMo4–5, 10CrMo9–10, X10CrMoVNb9–1) showed diversified morphology of the formed oxide layers. In the case of oxides on the inside, a trend of growing isotropy was shown with increasing operating temperature (an increase from 0.33 to 0.78). However, the layer of oxides formed on the outside of low-chromium steels was highly isotropic, while that formed on high-chromium steels was perfectly anisotropic [29].

The Pilling–Bedworth ratio is a well-acknowledged metric for evaluation of the compactness of oxide films formed on an alloy surface, as it accounts for the volume change when a passive film forms on a metal surface. According to the Pilling–Bedworth relationship, it was found that a more closely packed surface with a lower surface energy tends to exhibit superior corrosion resistance. In addition, surface energies have often been used in atomistic simulations to gain insights into corrosion performance of alloys [30]. 

In summary, improving the thermal fatigue resistance, oxidation resistance, and high-temperature friction and wear resistance of the mold steel are considered as effective ways to improve the accuracy and life of the mold. In addition, the initiation and expansion of fatigue cracks are closely related to oxidation. Therefore, the improvement in the toughness and oxidation resistance is beneficial to delaying the initiation and expansion of cracks during the service process of the mold, and leading to the increase in the service life of the mold. On the other hand, wear resistance is also the main failure mode of hot-working dies, especially extrusion molds and forging dies, and is also closely related to thermal properties and oxidation resistance. The wear resistance of the mold can be guaranteed under certain hardness conditions. On this basis, the improvement in the anti-oxidation performance can contribute to slowing down the formation of the oxide layer on the surface of the mold and oxidized abrasive particles, and reduce the stress field conditions of the mold, oxidative wear, and abrasive wear. At the same time, the increase in toughness is beneficial to preventing the plastic deformation of the subsurface layer of the hot-working mold under large stress load conditions, and can prevent further initiation and expansion of cracks in the mold matrix caused by plowing or abrasive particles penetrating into the matrix. Therefore, it can be concluded that improving the oxidation resistance and toughness is conducive to comprehensively improving the resistance to high-temperature thermal mechanical fatigue and high-temperature wear resistance of die steel. However, the current strengthening methods make it difficult to significantly improve the comprehensive performance of die steel while simultaneously meeting economic conditions. Therefore, there is an urgent need to develop new strengthening technologies that can significantly enhance the service performance of die steel and achieve industrialization.

## 3. Main Methods of Toughening Die Steel

Hot-work molds are subjected to a large number of local impacts, cyclic loading, thermal stresses, and corrosion, significantly affecting their service life. The main problems faced are wear of the mold geometry, plastic deformation or dimensional instability, and surface cracks. The life of molds depends upon various factors such as quality of the design, appropriate choice of materials, correct manufacturing, correct heat treatment, and selected working conditions. Therefore, choosing the appropriate die material is an important factor in successful product design [31]. It is reported that traditional die steels cannot adapt to the development of modern manufacturing and die steels require high strength and hardness, especially high heat intensity, high heat fatigue resistance, high toughness, and good abrasion resistance. To meet excellent properties of mold steel, the method of strengthening and toughening mold steel has become the research goal of scientists [32,33,34].

### 3.1. Optimizing Alloying Elements

In order to obtain superior thermal stability of hot-work die steel, the microstructure of hot-work die steel must be optimized by alloying (Cr, V, Mo, Si) [35] and process optimization (vacuum slag, ultra purification, and heat treatment). For these toughening methods, the strengthening effect and actual cost should also be considered when adding alloys. Mo and V elements are strong carbide-producing elements. Mo plays a profound role in the retardation of coarsening of nano-carbide and dislocation annihilation during tempering at high temperatures. The mechanical properties of steel alloys with Mo addition can still be maintained after high temperature aging. Conversely, V-rich MC particles can coarsen during hot rolling or subsequent annealing heat treatment, resulting in a significant decrease in material strength [36,37]. The element carbon is the most important alloying element in steel, and it presents in steel either from solids on the steel substrate or from participating in carbide formation, which can increase the strength and hardness of steel while decreasing toughness. Therefore, appropriate C content in the alloy can ensure sufficient hardness in a high-temperature environment and avoid severe toughness deterioration [38]. Moreover, Si has a significant effect on increasing the number and stability of residual austenite.

The comprehensive performance of mold steel is closely related to appropriate alloy composition. Du et al. [39] designed a new 5Cr5Mo2 mold steel by thermodynamic calculations using Thermo-Calc software based on the most common hot-work H13 die steel. The thermodynamic calculation results of the equilibrium precipitation phases of the two steels and the change curves of the constituent elements of the relevant phases are shown in Figure 1. The thermal stability test was performed at a temperature of 600 °C, and the calculated results for the selected temperature range of 400 to 1500 °C are shown in Figure 4a,b. As can be seen, the two steels have the same type of equilibrium precipitation phases, while the precipitation temperatures are different, which should be attributed to the transformation of the stable phases of the two steels after the tempering at 600 °C. The calculated results of the composition changes of these phases shown in Figure 4c–h indicate that the mole fractions of the four stable phases (α, MC, M_23_C_6_, M_2_C) of 5Cr5Mo2 steel at 600 °C are 90.0%, 0.4%, 8.7%, and 0.9%, respectively. In addition, the mole fractions of stable phases (α, MC, M_23_C_6_, M_6_C) in H13 steel are 94.2%, 4.5%, 1.1%, and 0.2%, respectively. These results are consistent with the expected alloy design.

Comparing with H13 steel, the Mo content of DIEVAR steel increased from 1.25 wt.% to 2.25 wt.%, while Si and V simultaneously decreased from 1 wt.% and 0.60 wt.% to 0.35 wt.% and 0.60 wt.% respectively. The addition of Mo lead to M_2_C-enhanced co-lattice deposition, and the reduction of Si altered secondary carbide deposition. It is proved that reducing V can inhibit the formation of VC eutectic carbide to a certain extent. DIEVAR steel has higher toughness, ductility, and fatigue strength than that of AISI H13 steel. However, the thermal crack length of DIEVAR steel after 2000 cycles is larger than that of high chromium martensite die steel and is less stable when tempered at 650 °C [40,41].

Xiang et al. [42] formed a new steel Cr5 (the content of V is 0.5–0.65%) based on DIEVAR (the content of V is 0.5–0.65%) and added 1% W to Cr5W steel to form W_2_C/WC. In addition, the effect of W on the hardness and toughness of Cr5 steel was studied. They found that W inhibited the coarsening of secondary M_6_C carbide and prevented the formation of M_23_C_6_ and M_7_C_3_. In addition, it can also improve red hardness by 6.3–15% after tempering at 600 °C for 10–40 h. The presence of a precipitated phase plays a crucial role in the improvement in the mechanical properties of steel.

In addition to common elements, rare earth (RE) elements are also widely used to refine grain size, change inclusions, and increase impact toughness. Cerium and lanthanum can produce extremely stable oxides, oxygen sulfide, and sulfide, which are beneficial in dispersing the carbide and improving the hardening of steel processing [43]. According to the first principle, the lanthanum atom has a strong affinity with carbon atom in four nearest neighbors and five coordination shells. In addition, the interaction between rare earth and carbon atoms can also influence the behavior of carbide precipitation [44]. Zhu et al. [45,46] found that the dislocation density in H13 molded steel increased by 205.0% with the addition of La. Therefore, dislocation strengthening and precipitate strengthening that occur under RE microalloying conditions improve the work-hardening ability of H13 die steel. In addition, the rare earth yttrium (Y) was also employed to improve the tensile properties and work-hardening ability of H13 die steels. This work can not only provide ideas for the design of high work-hardening ability alloy steel, but also lay a foundation for understanding the microalloying mechanism of rare earth elements [44,46,47].

Although the microstructure and performance of die steel can be improved by adding alloying elements, enhancing the strength and toughness of die steel becomes increasing challenging.

### 3.2. Electroslag Remelting

In addition to optimizing alloy composition, the microstructure and properties of steel can also be improved by improving the production process. Among these methods, electroslag remelting (ESR) is widely used in the production of high-quality tool steel in order to improve the cleanliness, solidification microstructure, and carbide size [48]. Li et al. [49] studied the effect of axial static magnetic field (ASMF) on carbides and mechanical properties of electroslag remelted (ESR) Cr12MoV die steel. Figure 5a–f show a schematic diagram of the flow pattern during the electromagnetic-controlled electroslag remelting (MC-ESR) process. The superposition of 30 mT ASMF causes unidirectional rotation in the molten slag pool, resulting in a more uniform temperature distribution at the front of the dendrite solid–liquid interface and smaller size of grains and carbides in Cr12MoV steel ingots. After the application of ASMF, the sulfur content of the ingots was decreased and the mechanical properties were improved. They believe that the unidirectional flow should be attributed to the interaction between dendrites in the melt caused by the coupled thermo-electromagnetic force inducing, which promotes the desulfurization process and the refinement of grains and carbides.

The current research focuses on the effect of electroslag remelting and directional solidification technology (ESR-CDS) on the production of high-temperature alloys. Qi et al. [50] demonstrated that the combination of directional solidification technology and electroslag remelting technology can effectively eliminate segregation behavior in ingots. Figure 5g–j show the cross-sectional view of the molten pool and the schematic diagram of the solidification behavior during different remelting methods. As can be seen, a shallower and flatter melt pool can be observed during the ESR-CDS process. High cooling rates with water spraying at the bottom of ingots can provide a stable temperature gradient almost parallel to the axis of the ingot [51]. ESR-CDS can effectively decrease the number and size of inclusions in the product by the shallow metal molten pool controlled by directional solidification. Meanwhile, the smaller secondary dendrite spacing and smaller non-equilibrium precipitation phases can be obtained by the higher cooling rate in the ESR-CDS process. Compared with the ESR process, the ESR-CDS process improves the distribution and morphology of carbides in the ingot and can obtain finer primary carbides and more dispersed carbides. This technology is beneficial to eliminating macro-segregation and ensuring less micro-segregation in the ingot to obtain better mechanical properties.

Electroslag remelting technology has become the current standard technology in the field of special steel, while adding electromagnetic control and directional solidification based on electroslag remelting technology cannot be applied in special steel production in a short time because it not only requires corresponding transformation of equipment and production processes, but also greatly increases the metallurgical cost of steel.

### 3.3. Increasing Forging Ratio

The hot forging process is often used to obtain a more compact and uniform microstructure of the die steel. As a plastic deformation process, a forging process can not only break the coarse and large casting microstructure into a fine fibrous microstructure and eliminate casting defects [52], but also realize dynamic recrystallization, and store a lot of energy and dislocation density with the amount of deformation [53,54,55]. In order to control the precision forging forming quality and prolong the service life of the steel, forging process parameters are optimized. In these parameters, the forging ratio has a significant effect on the microstructure performance of the forging. When the forging ratio (K) was less than 2.0, the bubbles, looseness, and microcracks in the ingots were partially welded together under compressive stress. The thicker dendritic core cylindrical microstructure was broken and recrystallized into fine grains, and some nonmetallic ingots fractured together with the ingots due to a concentration of carbide dispersion on the grain boundary. When forging ratio is within the range of 2.0–5.0, the carbide and metal flow form fibrous microstructure at the grain boundary, leading to directional properties of the forging. As the forging ratio increases, the longitudinal plastic index of the forging increases while the transverse plastic index decreases significantly. Consistent fibrous tissue forms in the forging when the forging ratio exceeds 5.0. Heat forging is a kind of high-temperature plastic deformation. The work-hardening effect and dynamic softening effect occur simultaneously during the thermal deformation of materials. Some static recrystallization occurs at a lower amount of deformation. As the amount of deformation increases, a great deal of deformation storage energy and lattice distortion energy are accumulated in the grain, resulting in dynamic recrystallization of the material and the formation of fine recrystallization grains. The sub-dynamic recrystallization phenomena occur in the existing dynamic recrystallization gap, and the grains still grow at a high temperature at the end of forging [56,57,58,59].

Zhou et al. [60] removed the non-metallic inclusions by means of refining, in which the initial forging temperature was controlled at 1200 °C and the final forging temperature exceeded 850 °C after the forging of 20 t slag ingots by a 60 MN forging machine. Z-direction upsetting and X-direction elongation were used to crush the microstructure of large branch crystal and column crystal. The ingot with a small cross-section can provide favorable conditions for the treatment of high-temperature homogenization in the future. Then, the direction of X-upsetting and Y-elongation achieved the uniform deformation due to the radial intersection of the two directions, the compaction of the core and the crystal branches, the complete crush of cylindrical crystal, and more uniform distribution of the carbide. Finally, Y-heading and Z-heading are realized, which completely solves the phenomenon of material segregation and improves material equidirectional performance. In short, a large forging ratio is conducive to obtaining high-performance special steel [61].

### 3.4. Heat Treatment Process Enhancement

It is generally believed that optimizing chemical composition, manufacturing method, and heat treatment process are effective ways to improve the properties of hot molded steel [55,56,57]. The heat treatment process includes deep cold treatment, spheroidal annealing, austenitic solidification, and quenching tempering (QT) [62,63]. Among these methods, QT is considered an excellent technique to improve mechanical properties of die steel by adjusting residual austenite (RA) volume fraction, martensite slab size, carbide shape, and dislocation. Hence, it is very important to use heat treatment effectively [64,65].

Yu et al. [65] investigated the effects of quenching temperature and tempering temperature on microstructure evolution and mechanical properties of 55NiCrMoV7 hot-molded steel. Typical tempering microstructures composed of tempered martensite, fine needle-shaped precipitated carbide, quasi-spherical insoluble carbide, and residual austenite were obtained after tempering for 5 h at 790–910 °C and tempering for 5 h at 600 °C. An obvious coarsening of the grains can be observed with the increase in the quenching temperature, and the grain structure develops towards an equiaxed shape, which leads to the enlargement of the tempered martensite size. As carbon and other alloying elements such as Cr, Mo, and Mn may dissolve into the austenite phase, the volume fraction of retained austenite increases. Both tempered martensite and precipitated carbides are sensitive to tempering temperature. The lath width of tempered martensite increases with increasing the tempering temperature [66] and the precipitated carbides are not suitable for nucleation at low temperatures. After the temperature rise, the carbides undergo a transformation from ɛ-M_23_C_6_-spherical M_23_C_6_-short rod-shaped M_7_C_3_. Finally, as the tempering temperature continues to increase, only the coarsening of M_7_C_3_ carbides occurs [62,67].

Zhou et al. [68] found that increasing tempering temperature promotes the precipitation and coarsening of spherical and needle-like Mo_2_C carbide in DM hot molds. Cheng et al. [69] reported that the carbide was separated with increasing tempering temperature and showed a chain distribution. Meanwhile, the tensile strength increased while the yield strength, impact toughness, and fracture toughness decreased. Liu et al. [70] showed that the yield and tensile strength decreased but the impact toughness increased with increasing tempering temperature, which may be due to the evolution of precipitated carbide (PC), martensite slats, and dislocations. Therefore, an appropriate heat treatment process is required to obtain the appropriate organization and further improve properties [71,72,73].

## 4. Nanoparticle Strengthening Technology

### 4.1. Research Status of Nanoparticle Strengthening Technology

The steel industry is continuously looking for new ways to improve the comprehensive performance of steel while simultaneously meeting environmental protection, and economic and other conditions. A new nanoparticle enhancement technology has attracted the attention of scholars in recent years and has been researched and applied extensively. Nanoparticles can not only interact with dislocations, but also serve as the core of grains to promote heterogeneous nucleation of grains and grain refinement. Whether as nucleation phase or strengthening phase, the lattice mismatch and valence electrons between nanoparticles and the main target phase of steel have been believed to affect the nucleation efficiency and strengthening effect. Nanoparticles are found to be capable of notably optimizing the nucleation and precipitation behavior of steel, and ultimately regulating the microstructure and properties of steel, because nanoparticles possess a coherent/semi-coherent relationship with the nucleation phase of steel and can serve as the core of the steel nucleation. In particular, when the lattice mismatch energy between the nanoparticles and the nucleation phase of the steel is less than 12%, the heterogeneous nucleation effect is significant [74,75,76,77,78,79,80,81,82,83].

At present, additive methods and endogenous methods are commonly used to introduce nanoparticles into steel. Compared with the endogenous method, the external method can simultaneously control the size and proportion of nanoparticles. However, nanoparticles are difficult to uniformly disperse in steel through external addition because nanoparticles with high surface energy, low wettability, and density difference between nanoparticles and steel agglomerate easily and float on the surface of molten steel [84].

Some scholars have tried different methods to disperse nanoparticles in the molten steel evenly by modifying the surface of nanoparticles [85,86]. Qin et al. [87] successfully prepared nano-NbC particle-reinforced case low-carbon steel by mixing nano-NbC particles with Fe power by mechanical alloying technology. It was found that the nano-NbC particles can achieve homogeneous dispersion in the melt (Figure 6a–d).

Additive manufacturing allows the synthesis of complex geometries and optimizes lightweight design and enhanced functionality [88]. Nanoparticles are effective sites for heterogeneous nucleation at the solidification front during this process, promoting the fine equiaxed grain microstructure and simultaneously achieving fine grain strengthening and dispersion strengthening. The technology can be divided into two parts as follows: the synthesis of nanoparticles and metal powders through various methods such as inert gas atomization and mechanical grinding and the selectively melt of a layer of specific powder material by a selective laser. Figure 6e,f show the SLM device system and laser scanning strategy, respectively. Three main problems need to be solved for laser-based additive manufacturing technology, according to previous research. Firstly, the added heterogeneous particles (diameter ≤ 5 μm) can reduce the fluidity of the matrix powder. Secondly, the melting and solidification process is complicated due to the different heat input between the heterogeneous particles and the metal caused by different laser absorption rate. Thirdly, it is difficult to control the wettability of the metal matrix and heterogeneous particles [89,90,91].

Tang et al. [92] successfully achieved the formation of nanoparticles in the metal melt by a combination of multi-point dispersion supply processing and electromagnetic stirring (Figure 6g). The multi-point dispersion supply technology is a new method to generate nanoparticles in liquid melts by controlling the solute content at the front of the interface. First, the pure titanium wire added to the melt is multi-point-dispersed and the free oxygen concentration should be controlled within tens of ppm. Secondly, a convection field is formed in the melt through electromagnetic stirring or Ar bottom blowing, which promotes the flow of molten metal. Titanium will react with oxygen and a large number of homogeneous nucleation particles of titanium oxide are formed during this process. Finally, the melt is poured into the casting mold at air-cooling speed and the nanoparticles are evenly dispersed within the cast steel ingot. This method has been used in the subsequent development of high-strength low alloys and weld metals to effectively improve microstructure and properties by optimizing inclusions, refining grains, and increasing strength to resist stress corrosion cracking without losing plasticity [93].

Qiu et al. [94] used a master alloy method to obtain uniformly distributed nanoparticle-reinforced steel and the specific process is as follows. First, the master alloy containing nanoparticles was prepared through self-propagating high-temperature synthesis, stirring casting, and powder pressing. Secondly, the metal melt was poured into a pouring ladle containing the master alloy or the master alloy was added to the molten metal and then stirred (as shown in Figure 6h). Finally, the nanoparticles were evenly dispersed in the metal matrix during the tumbling process of the molten metal (as shown in Figure 6i). The intermediate alloys hinder the uneven dispersion and segregation caused by the different specific gravity between nanoparticles and matrix. This method refines the microstructure of the die steel and improves the strength and toughness of the steel at the same time as well as achieving a good combination of endogenous methods and external methods. Its application can be conducive to large-scale or industrial-scale production.

**Figure 6 materials-17-03455-f006:**
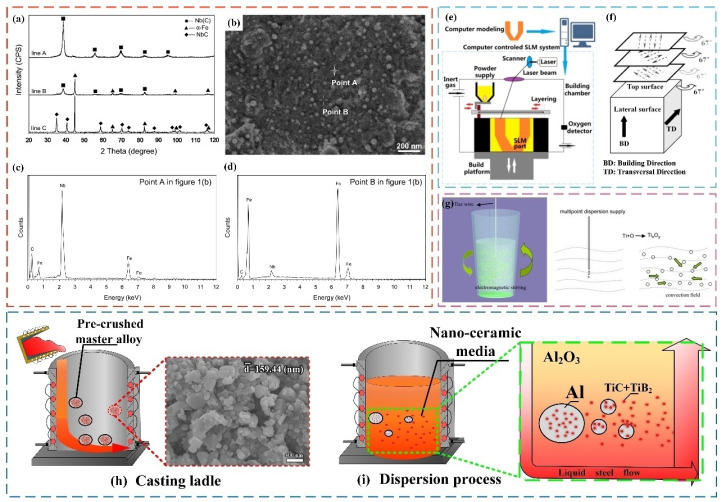
(**a**) X-ray diffractograms of the mixture before adding Fe powder, after adding Fe powder, and after heat treatment. (**b**) SEM image of nanopowder morphology. EDX analysis of different substances in (**b**): (**c**) particles; (**d**) matrix [87]. Schematic diagram of SLM equipment: (**e**) SLM device system; (**f**) laser scanning strategy [89]. (**g**) Principle diagram of the technology combining multi-point-dispersed supply processing and electromagnetic stirring [92]. Flow chart of using master alloy to prepare nanoparticle reinforced steel: (**h**) casting ladle; (**i**) dispersion process [94].

### 4.2. Effect of Nanoparticles on Microstructure of Steel and Its Mechanism

Nanoparticles have a positive effect on the microstructure of steel by the dispersion strengthening, and the microstructural evolution can seriously affect the mechanical properties of steel. Many scholars have conducted extensive research on the effect of nanoparticles on the microstructure of steel [95,96,97,98,99]. For example, Park et al. [100] prepared TiC-reinforced SA106B steel by conventional powder metallurgy. They studied the effect of nano-TiC on the casting microstructure of SA106B carbon steel. The equiaxed crystals usually appear at the center, while columnar crystals are produced outside (as shown in Figure 7a). Compared with the ingot without the addition of nanoparticles, the ingot has a wider equiaxed crystal microstructure area and a narrower columnar crystal area (as shown in Figure 7b). This is because the added nano-TiC acts as a heterogeneous nucleation particle during the nucleation process, accelerating the nucleation and growth of equiaxed crystals, and inhibiting the growth of columnar crystals.

Wang et al. [93] prepared low-activity ferrite/martensitic steel (RAFM) containing ZrC with a size of 30 nm by plasma sintering and hot-rolling methods and studied the influence of nanoparticle content on the microstructure of steel. They found that the addition of ZrC with different content has a significant grain refinement of the test steel. The addition of 0.75 wt.% ZrC especially had the best refinement effect on the microscopic combination of the test steel (Figure 7c–g).

Researchers have also found that the added nanoparticles not only have a significant impact on the matrix microstructure, but also have a significant impact on the precipitated phase. Zhu et al. [101] successfully added nanoceramic additives to 310S austenitic stainless steel. Figure 7h,i are schematic diagrams of the microstructure evolution of 310S austenitic stainless steel during the solidification process without nanoceramic additives and with nanoceramic additives, respectively. Compared with the 310S austenitic stainless steel without nanoceramic additives, the initial precipitated phase M_23_C_6_ was observed in the 310S austenitic stainless steel with nanoceramic additives. This is because nanoceramics can acted as the nucleation particles of the primary large-volume M_23_C_6_. In addition, layered M_23_C_6_ carbides and spherical M_23_C_6_ carbides were generated around the bulk M_23_C_6_ carbide during further solidification process.

Hong et al. [102] successfully added TiC powder to the molten steel by a mechanically activated sintering process to explore the effect of nanoparticles on the microstructure of steel. They compared the microstructure of SA-106B carbon steel with no TiC and 0.1 wt.% TiC addition after casting, hot-rolling, and normalizing processes, respectively, as shown in Figure 7j,k. This figure reveals the presence of ferrite and pearlite for nanoparticle-strengthened carbon steel after being subjected to different thermomechanical routes. It can be speculated that the growth rate of recrystallized austenite during hot rolling and normalizing is strongly inhibited by TiC particles [103]. Obviously, this ultimately promotes the grain refinement of the ferrite–pearlite microstructure in carbon steel.

To sum up, nanoparticles are found to be capable of notably optimizing the nucleation behavior and precipitation process, improving the overall properties of steel. Therefore, it is also necessary to discuss the influence of nanoparticles on the mechanical properties of steel [98].

### 4.3. Nanoparticle Reinforcement of Service Performance of Steel and Its Reinforcement Mechanism

The microstructure of steel is crucial because it directly determines the mechanical properties of steel. On the one hand, the uniform dispersion of nanoparticles in steel can significantly refine the microstructure and precipitation phase of steel. On the other hand, nanoparticles can regulate the microstructure of steel during almost the entire casting, heat treatment, and processing process. Therefore, many researchers have explored the impact of nanoparticles on the mechanical properties of steel.

#### 4.3.1. Tensile Properties and Reinforcing Mechanism of Nanoparticle-Reinforced Steel

Zhai et al. [104] successfully prepared nanoparticle-reinforced 316 L stainless steel through selective laser melting and tested the tensile properties of 316 L stainless steel reinforced with different contents of TiC particles (Figure 8a,b). It can be seen that the addition of 1 wt.% and 3 wt.% TiC nanoparticles can significantly increase the yield strength and tensile strength while maintaining a high elongation. The yield strength of 316 L stainless steel increases from 608 MPa to 694 MPa and 773 MPa, respectively, and the tensile strength increases from 722 MPa to 888 MPa and 988 MPa, respectively. The true stress–strain curve shows that the work hardening rate of 316 L stainless steel increases after adding nano-TiC. The strengthening mechanism is mainly that TiC act as heterogeneous nucleation particles to refine the grains and TiC nanoparticles hinder the movement of dislocations. In addition, the decrease in elongation is due to the suppression of the twinning-induced plasticity effect by nano-TiC.

The high-temperature performance of mold steel deserves more attention because they are often used in high-temperature and high-pressure environments. Peng et al. [105] successfully prepared nanoparticle-reinforced austenitic stainless steel based on Super304H steel through mechanical alloying and hot isostatic pressing processes. They tested the tensile properties of steel at room temperature and high temperature (shown in Figure 8c), and compared their strength with commercial Super304H steel (shown in Figure 8d,e). The results show that nanoparticle-reinforced austenitic steel exhibits a high strength at room temperature and the tensile strength and yield strength reach 995 MPa and 700 MPa, respectively. The high-temperature tensile strength and yield strength of austenitic oxidation dispersion-strengthened heat-resistant steel are 94% and 110% higher than that of commercial Super304H austenitic steel, respectively. The calculation results indicate that Y_2_O_3_ exhibits a semi-coherent relationship with the austenite matrix. In addition, a coherent interface between the copper-rich phase and austenite matrix is formed and the Cu-rich phase is wrapped by Y_2_O_3_. The existence of a Cu-rich phase provides more nucleation sites for oxides, which effectively promotes the precipitation and diffusion distribution of oxides. The dispersed particles pin the movement of dislocations. The strengthening mechanisms mainly include dispersion strengthening, semi-coherent strengthening, and fine-grain strengthening.

Nanoparticles have different effects on the tensile properties of the matrix steel at different temperatures. Li et al. [106] studied the effect of in situ micron TiC particles on the tensile and fracture mechanism of martensitic wear-resistant steel at different temperatures. The results shown in Figure 8f–h indicate that the strength of TiC-reinforced wear-resistant steel is higher than that of conventional wear-resistant steel at each test temperature. Within 25–500 °C, there is a good connection between the TiC particles and the matrix and the TiC particles can bear stress well. As the tensile stress increases, the fracture will occur due to the formation of holes in the matrix. Above 500 °C, interface debonding between particles and matrix is the main mode of crack initiation. TiC ceramic particles can enhance high-temperature strength and reduce the elongation of the test steel. At 600 °C, the connectivity between the particles and the matrix becomes poor and voids are easily formed between the particles and the matrix, resulting in a significant reduction in the elongation of the particle-reinforced steel.

To sum up, ceramic particles can effectively enhance the yield strength and tensile strength of steel at room temperature and high temperature. However, in some cases, the addition of nanoparticles may result in a decrease in elongation. This may be related to the type, content, and size of the nanoparticles, or the type of matrix steel, experimental conditions, etc.

#### 4.3.2. Fatigue Performance and Strengthening Mechanism of Nanoparticle-Reinforced Steel

Fatigue is one of the most common failure modes of steel, which seriously affects its service life. Fatigue cracks are an intuitive means for evaluating the thermal and cold fatigue resistance of materials. Qiu et al. [94] successfully prepared a trace amount of TiC-TiB_2_ nanoparticle aluminum-based master alloy by introducing the nanoparticles into a new type of high-chromium hot-work die steel in the form of a master alloy and achieved uniform dispersion of the nanoparticles. Figure 9a,b show the schematic diagrams of the cracks in alloys without nanoparticles and with the addition of 0.02 wt.% TiC+TiB_2_ nanoparticles after 3000 cycles in the range of 650 °C–20 °C. It can be seen that the addition of a trace number of dual-phase nanoparticles significantly reduces the length and width of main crack in the test steel. Compared with the surface of the steel sample without nanoparticles, the number of cracks on the surface of the nanoparticle-enhanced steel is significantly reduced. It is confirmed that the addition of nanoparticles can effectively improve the thermal and cold fatigue resistance of steel and can refine grains and precipitated phases, which can contribute to the increased matrix strength and toughness and the reduced stress concentration.

In addition, repairing fatigue cracks is also an effective means to increase the service life of steel. Wang et al. [107] investigated the influence of nano-WC particles on the fatigue crack repair of 304 stainless steel. The probabilistic S–N curves (Figure 9c,d) indicate that the fatigue life of the repaired specimen can be significantly improved by adding WC nanoparticles, particularly in high-cycle fatigue environments. Therefore, the incorporation of nanoparticles shows a high potential for increasing the service life of the material during the fatigue repair process. 

#### 4.3.3. Wear Properties and Strengthening Mechanism of Nanoparticle-Reinforced Steel

The improvement in the wear and friction performance of steel usually depends on the microstructure and hardness of steel. However, the increase in the hardness alone cannot improve the wear performance of steel because of the cracks and peeling of steel caused by high hardness. Improving the hardness of the steel while retaining its toughness can prevent the microstructure of the steel from falling off surface during the wear process. Therefore, simultaneously increasing the hardness and toughness is vital for improving the wear resistance of steel. Some researchers have proved that a hard phase on the steel surface prepared by surface treatment methods such as surface shot peening, surface laser treatment, and surface spraying can increase the hardness [108,109]. However, the matrix still maintains a soft phase, and peeling or even deformation may occur under long-term load. An innovative approach has, therefore, also been proposed to enhance the wear resistance of steel by adding particles to form soft and hard phases [110,111,112].

Zhang et al. [113] investigated the synergistic effect of nano-TiC and retained austenite on impact abrasive wear behavior of bainitic steel. Figure 10a,b show SEM microscopic images of the longitudinal section of the worn surface of commercial wear-resistant steel (Hardox450) and bainite wear-resistant steel (BS), respectively. It can be seen that the cracks on the wear surface of Hardox450 have significantly expanded, while there are no obvious cracks on the worn surface of BS. This is due to the fact that TiC particles evenly distributed in the matrix can resist the intrusion of abrasives and the phase transformation of retained austenite improves the plasticity and alleviates the stress concentration caused by particles. Therefore, the synergistic strengthening of TiC and retained austenite effectively improves the impact wear resistance of bainite.

Shan et al. [114] studied the wear properties of high-manganese steel reinforced with micron-scale and nano-scale V_2_C carbides. A high-manganese steel strengthened with two sizes of particles was obtained by different heat treatment processes and the wear mechanism diagrams of two different strengthened steels are shown in Figure 10c,d, respectively. The micron-sized particles can effectively protect the matrix from wear in the early stages of wear. However, large-sized particles are prone to cracks and more likely to fall off the surface, which will seriously deteriorate the wear resistance of the test steel. In contrast, a large number of nanoscale particles with smaller sizes are evenly distributed in the matrix, which can effectively improve the wear resistance of test steel and provide support for micron-sized particles. Micron-sized particles that are more difficult to shed further enhance the wear resistance of the test steel.

The high-temperature wear of steel can be mainly divided into adhesive wear, abrasive wear, and fatigue wear. Abrasive wear is dependent on the size of the shed oxides. The bonding strength between the nano-sized particles and the matrix is higher than that between micron-sized particles and matrix. The micron-sized particles fall off more easily, which results in serious abrasive wear of the matrix. However, nanoscale particles can simultaneously enhance the strength and plasticity of the steel, and have good effects in enhancing resistance to adhesive wear, abrasive wear, and fatigue wear [113,115,116,117].

#### 4.3.4. Antioxidant Properties and Strengthening Mechanism of Steel Reinforced by Nanoparticles

The fatigue and wear behavior of steel are usually accompanied by oxidation behavior. Therefore, it is crucial to investigate the oxidation behavior of steel. For example, Kaito et al. [118] studied the effect of Y_2_O_3_ nanoparticles on the oxidation properties of 9Cr martensitic steel and 12Cr ferritic steel. They compared the anti-oxidation properties of the two kinds of nanoparticle-reinforced steels with the steels with different Cr contents (Figure 11a). The results indicate that the two nanoparticle-reinforced steels have better high-temperature oxidation resistance than that of the steels with Cr contents of 11 mass% and 17 mass%. Wu et al. [119] studied the influence of different TiC contents on the oxidation resistance of 304 stainless steel (304SS). Figure 11b,c show the weight changes in 304SS steel with different TiC contents after high-temperature oxidation. The results show that the addition of TiC particles significantly enhances the oxidation resistance of the test steel and 304SS–2TiC has the best strengthening effect. The addition of TiC particles maintained a low oxidation rate after oxidation at high temperature for 96 h and resulted in an improvement in the anti-flaking performance of the steel. This is attributed to the refinement of the grains and the increase in the diffusion rate of the Cr element caused by the added particles. In addition, the formation of the oxide film on the surface of TiC particles significantly enhances the oxidation resistance of 304SS. However, an appropriate number of particles should be added to achieve the best strengthening effect.

The content of Cr is positively correlated with the oxidation resistance of steel. Within a certain range, the higher the Cr content, the stronger the oxidation resistance of the steel. It is generally believed that the addition of nanoparticles results in the more uniform distribution of elements. During the oxidation process, nanoparticles can also affect the diffusion rate of different atoms, thereby affecting the oxidation performance of steel. The holes do not form easily in the steel because of the strong stability of nanoparticles. Moreover, the addition of nanoparticles can improve the oxidation resistance of the steel due to the pinning effect. 

## 5. Summary and Prospects

Hot-work die steel are essential materials for basic equipment in advanced manufacturing industries. Especially in recent years, with the rapid development of lightweight and integrated die-casting in the automotive industry, higher requirements are required for high-quality hot-work die steel. Due to the harsh service environment of high temperatures and pressure and hot–cold alternation, the life of the mold and the molding quality of the product are often seriously influenced by the three failure modes of hot-work die steel including fatigue failure, wear failure, and oxidation failure. Therefore, improving the resistance of these three aspects of die steel is the key to prolonging the service life of die steel. Traditional strengthening methods, such as alloy elements and heat treatment optimization, electroslag remelting, and a large forging ratio find it difficult to significantly improve the overall performance of mold steel or achieve industrialization. Therefore, there is an urgent need for developing new and efficient technologies for strengthening mold steel.

In recent years, nanoparticle strengthening technology has attracted much attention as a new technology that can effectively enhance the strength and toughness of steel. This article reviews the current research status of nanoparticle reinforcement technology and its roles and mechanisms in different aspects. In summary, the addition of nanoparticles can significantly refine the microstructure and precipitated phases of the matrix, resulting in excellent anti-fatigue, anti-wear, and anti-oxidation properties. However, nanoparticles are prone to agglomeration and difficult to disperse. Therefore, it is necessary to choose an adding method to maximize the overall performance while ensuring uniform dispersion of nanoparticles. Depending on the type of base steel, the size, type, and content of the selected nanoparticles are also different. Combining the external additive method with the endogenous method using an aluminum-based master alloy as the carrier of nanoparticles to achieve the addition of nanoparticles is the most outstanding innovative method in the field of nanoparticle-reinforced hot-work die steel. On the one hand, this method has significantly improved the comprehensive performance of hot-work die steel. On the other hand, the reinforced steels prepared by this method are also highly economical because of the trace addition of particles and the fact that they have been successfully industrialized. The nano-strengthened hot-work die steel technology is a promising method in steel fields.

It is hoped that this review can provide future researchers with a new idea to optimize particle content, size, and type for different steel, and even develop new strengthening technologies to meet new requirements for the comprehensive performance of steel in the new era.

## Figures and Tables

**Figure 1 materials-17-03455-f001:**
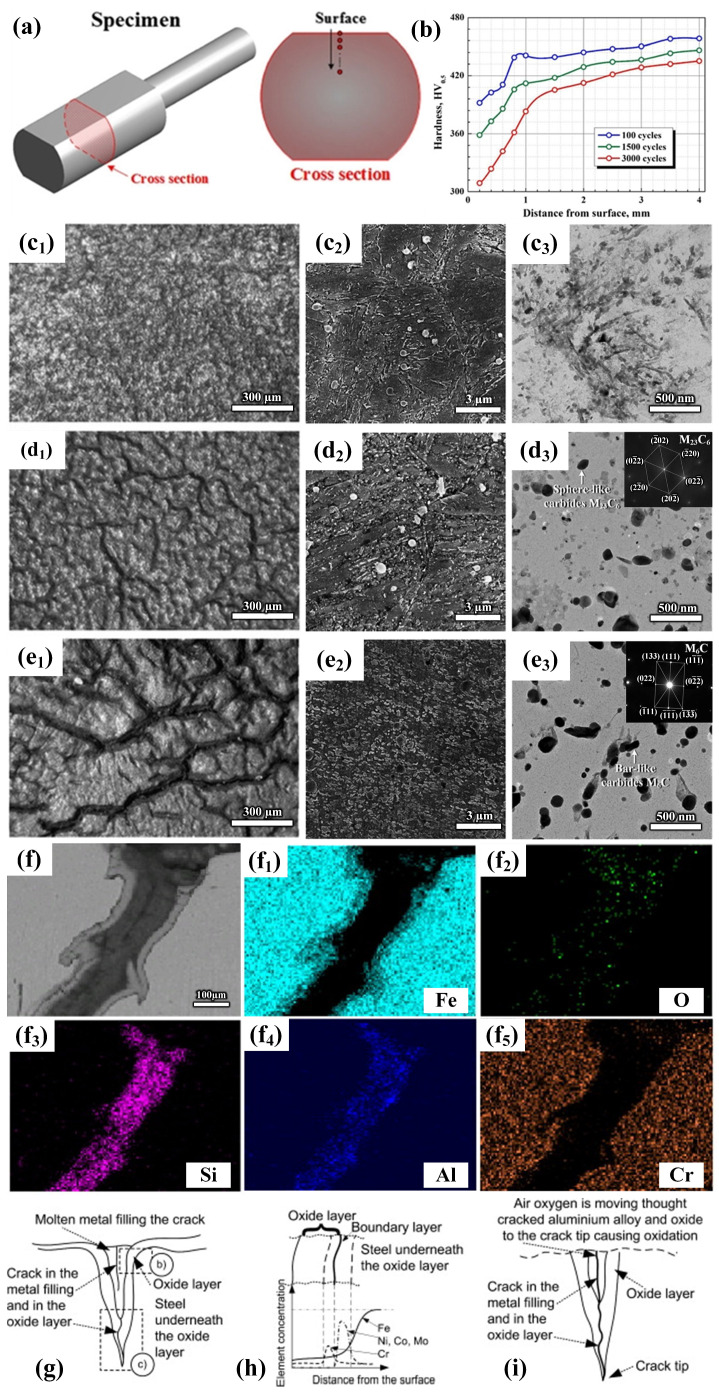
(**a**) Schematic diagram of experimental sample. (**b**) Hardness change curve after different thermal cycles. Macroscopic morphology of surface cracks after different thermal cycles: (**c_1_**) 100 times; (**d_1_**) 1500 times; (**e_1_**) 3000 times. SEM morphology images of surface cracks after different thermal cycles: (**c_2_**) 100 times; (**d_2_**) 1500 times; (**e_2_**) 3000 times. TEM morphology images of surface cracks after different thermal cycles: (**c_3_**) 100 times; (**d_3_**) 1500 times; (**e_3_**) 3000 times [9]. (**f**) SEM image of thermal fatigue cracks in the test steel after 20,000 cycles of immersion testing. Distribution maps of selected alloying elements around thermal fatigue cracks: (**f_1_**) Fe; (**f_2_**) O; (**f_3_**) Si; (**f_4_**) Al; (**f_5_**) Cr. (**g**–**i**) Schematic diagram of the mechanism of the effect of oxidation on hot crack growth [11].

**Figure 2 materials-17-03455-f002:**
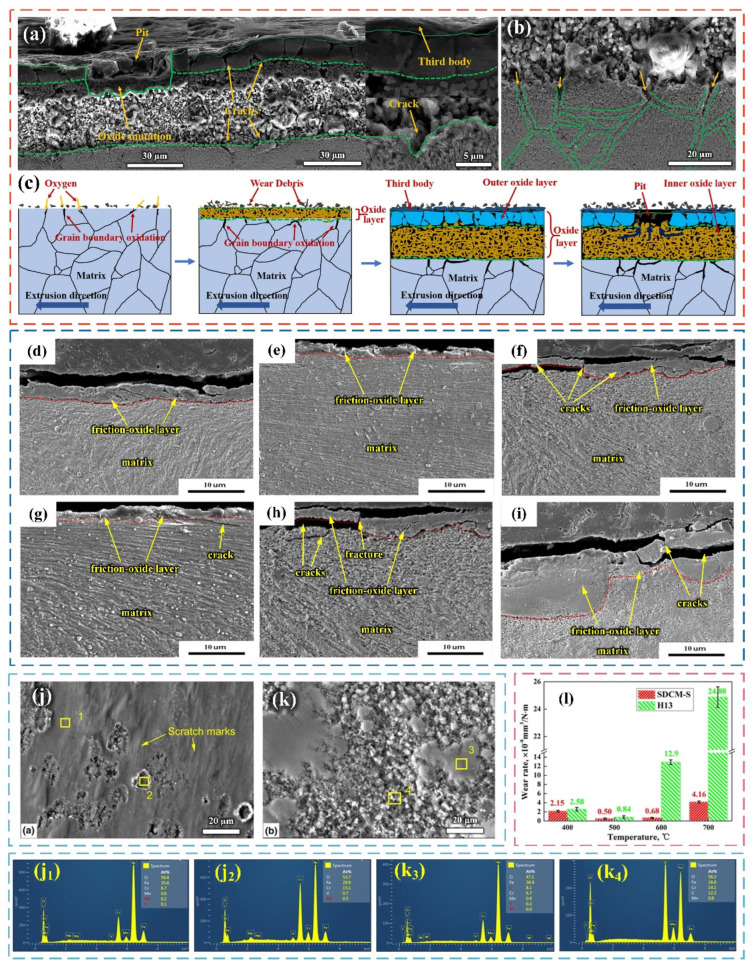
Microscopic image of the longitudinal section of H13 die steel after hot extrusion: (**a**) oxide layer; (**b**) metallographic microstructure; (**c**) schematic diagram of the mechanism of oxidation wear [19]. Cross-sectional morphology of H21 steel under different conditions: (**d**) 150 N, 400 °C, and 50 r min^−1^; (**e**) 150 N, 400 °C, and 100 r min^−1^; (**f**) 150 N, 500 °C, and 50 r min^−1^; (**g**) 150 N, 500 °C, and 100 r min^−1^; (**h**) 150 N, 600 °C, and 50 r min^−1^; (**i**) 150N, 600 °C, and 100 r min^−1^ [20]. SEM topography and EDS images of the worn surface of high-speed steel pins on prefabricated oxidized disks at different temperatures: (**j**) 850 °C; (**k**) 900 °C [21]. (**l**) Histogram of wear rate of SDCM-S steel and H13 steel at 400–700 °C [22].

**Figure 3 materials-17-03455-f003:**
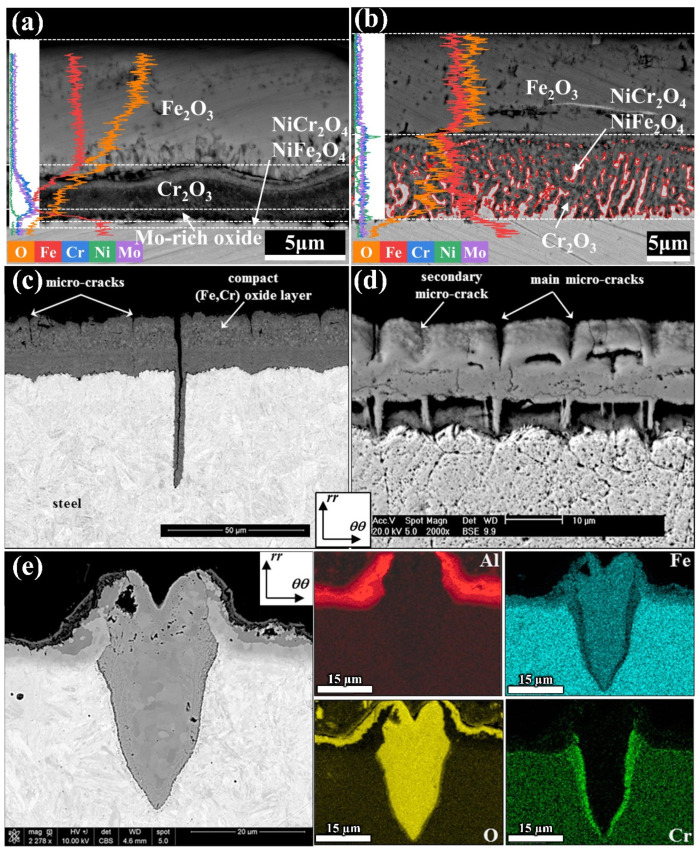
SEM–BSE and SEM–EDS line scan images of different steel types: (**a**) 3Cr3Mo2NiW steel; (**b**) 3CrNi3Mo steel [23]. Cross-sectional SEM images of mold steel specimens after 30,000 thermal fatigue cycles: (**c**) before electrolytic etching; (**d**) after electrolytic etching. (**e**) Cross-sectional scanning electron microscope image of microcracks formed on a coated sample after 30,000 thermal fatigue cycles, and SEM images of Al, Fe, O, and Cr distributions [28].

**Figure 4 materials-17-03455-f004:**
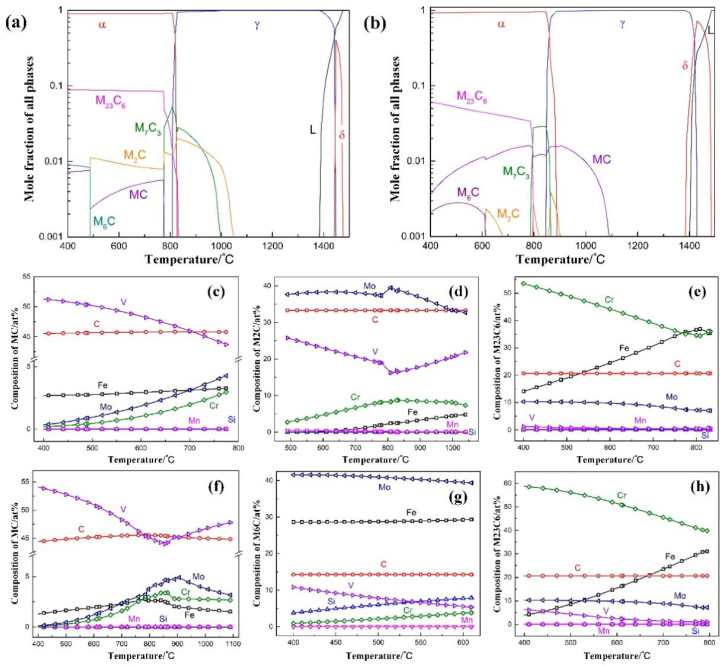
The stability of the phase with increasing temperature under equilibrium conditions: (**a**) 5Cr5Mo2 steel; (**b**) H13 steel. Elemental change curve of related phases of the 5Cr5Mo2 steel: (**c**) MC; (**d**) M_2_C; (**e**) M_23_C_6_; and the H13 steel: (**f**) MC; (**g**) M_6_C; (**h**) M_23_C_6_ [39].

**Figure 5 materials-17-03455-f005:**
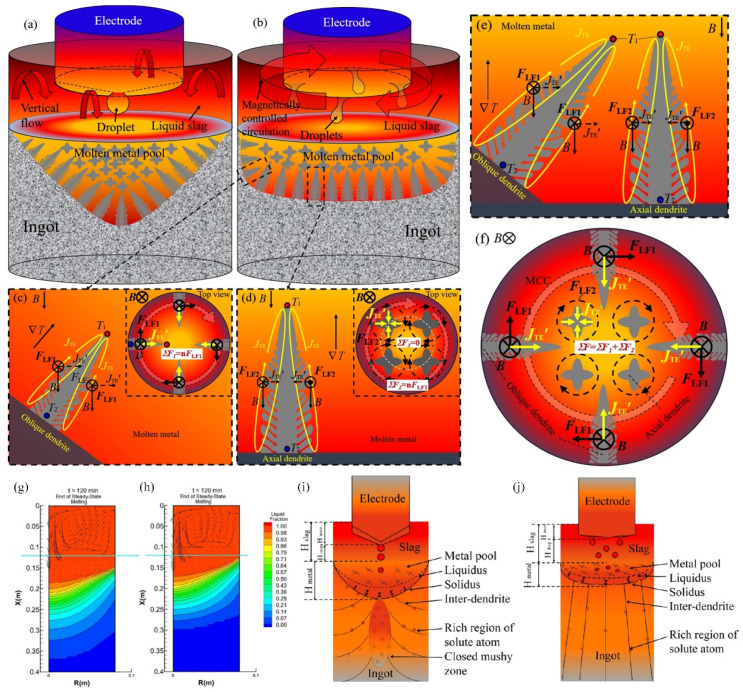
Schematic diagram of the mechanism of different remelting processes: (**a**) ESR; (**b**) MC-ESR. The effect of ASMF on different materials during the remelting process: (**c**) oblique dendrites; (**d**) axial dendrites; (**e**) molten pool; (**f**) top view [49]. Molten pool profiles during different remelting processes: (**g**) ESR; (**h**) ESR-CDS. Schematic diagram of solidification behavior during different remelting processes: (**i**) ESR; (**j**) ESR-CDS [50].

**Figure 7 materials-17-03455-f007:**
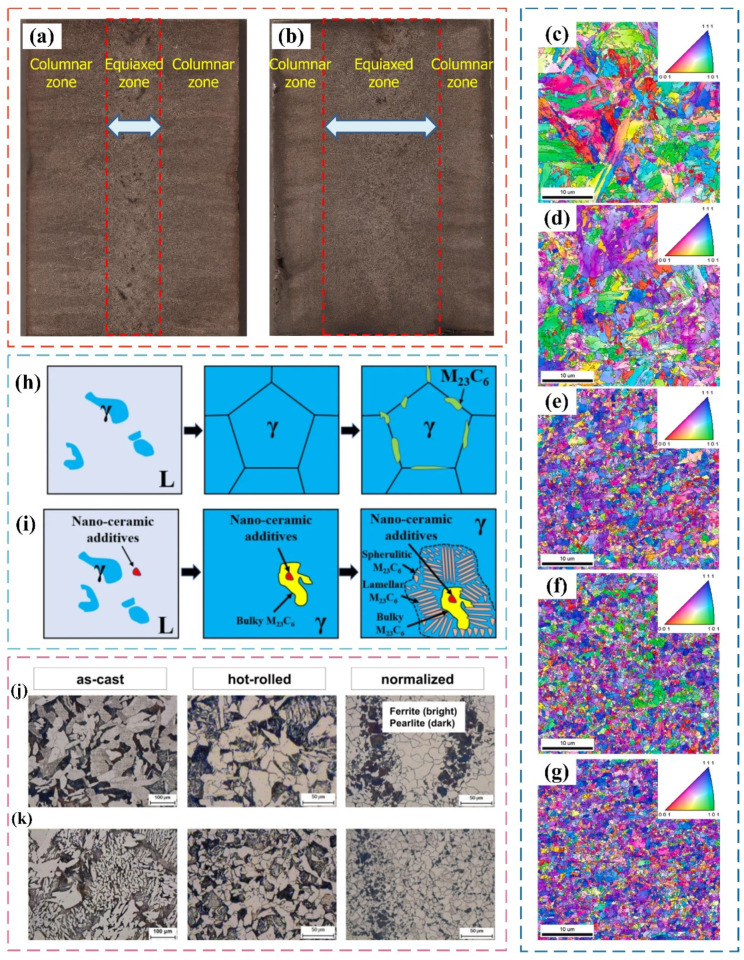
Macroscopic images of SA 106B carbon steel: (**a**) without nano-TiC particle addition; (**b**) with nano-TiC particle addition [100]. Inverse pole figure (IPF) map of RAFM steel with different ZrC contents: (**c**) 0 wt.%; (**d**) 0.25 wt.%; (**e**) 0.5 wt.%; (**f**) 0.75 wt.%; (**g**) 1.0 wt.% [93]. Schematic diagram of the microstructure evolution of 310S stainless steel during the solidification process: (**h**) without nanoceramic particle additives; (**i**) with nanoceramic particle additives [101]. Schematic diagram of the microstructure of SA-106B carbon steel after casting, hot rolling, and normalizing: (**j**) without adding TiC; (**k**) adding 0.1 wt% TiC [102].

**Figure 8 materials-17-03455-f008:**
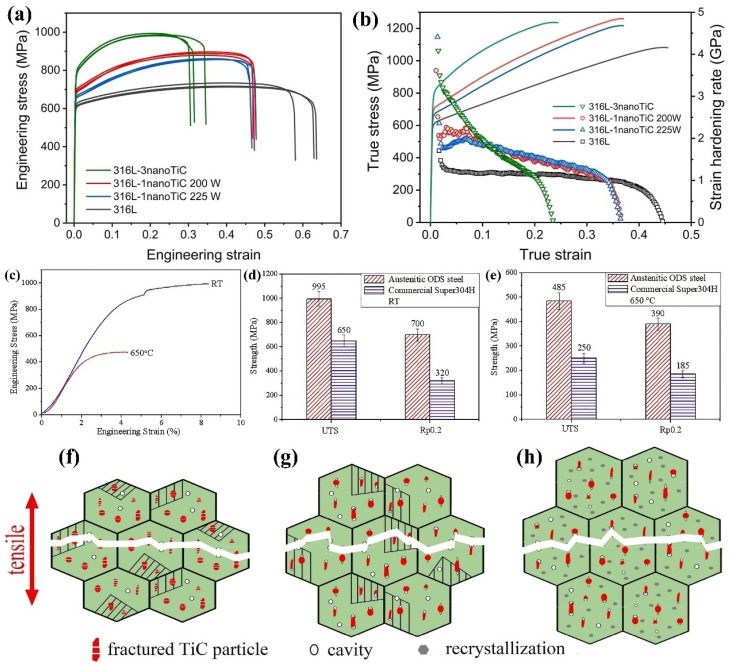
Stress–strain curves of stainless steel with different contents of nano-TiC-316L: (**a**) engineering stress–strain curve; (**b**) real stress–strain curve [104]. (**c**) Strain–stress curve of austenitic steel at room temperature and 650 °C. Comparison of the strength of Y_2_O_3_-reinforced austenitic heat-resistant steel and commercial Super304H steel at different temperatures: (**d**) room temperature; (**e**) 650 °C [105]. Schematic diagram of fracture of TiC-reinforced high-temperature martensitic steel at different temperatures: (**f**) 25–500 °C; (**g**) 500–600 °C; (**h**) >600 °C [106].

**Figure 9 materials-17-03455-f009:**
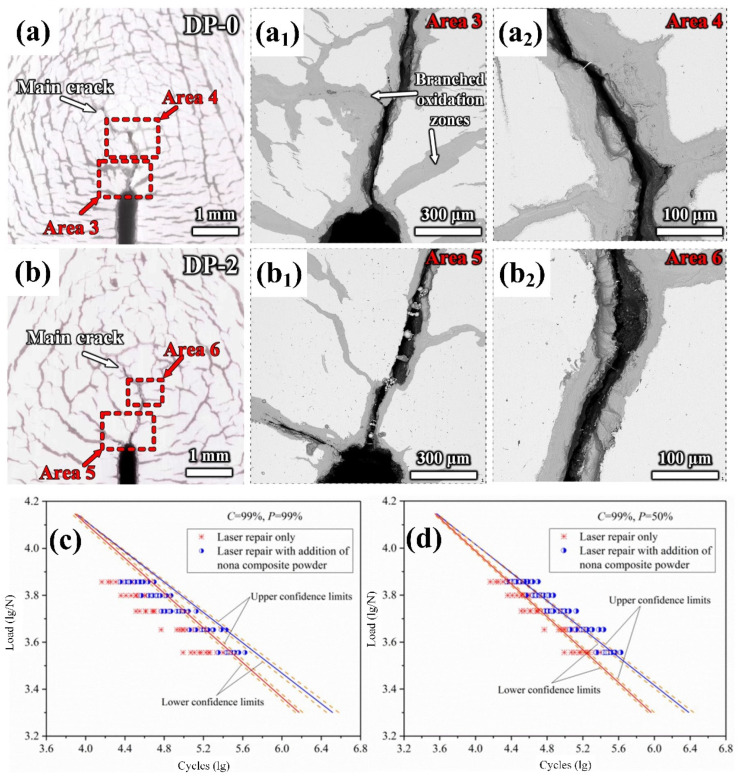
Microscopic diagram of thermal fatigue cracks at the preset notch tip of the new high Cr martensitic mold steel after 3000 thermal fatigue cycles at 650 °C to 20 °C: (**a**) without adding nanoparticles; (**b**) adding 0.02 wt.% TiC+TiB_2_ [94]. Probability S–N curve at the specified level: (**c**) confidence (C) = 99%, probability (P) = 99%; (**d**) C = 99%, P = 50% [107].

**Figure 10 materials-17-03455-f010:**
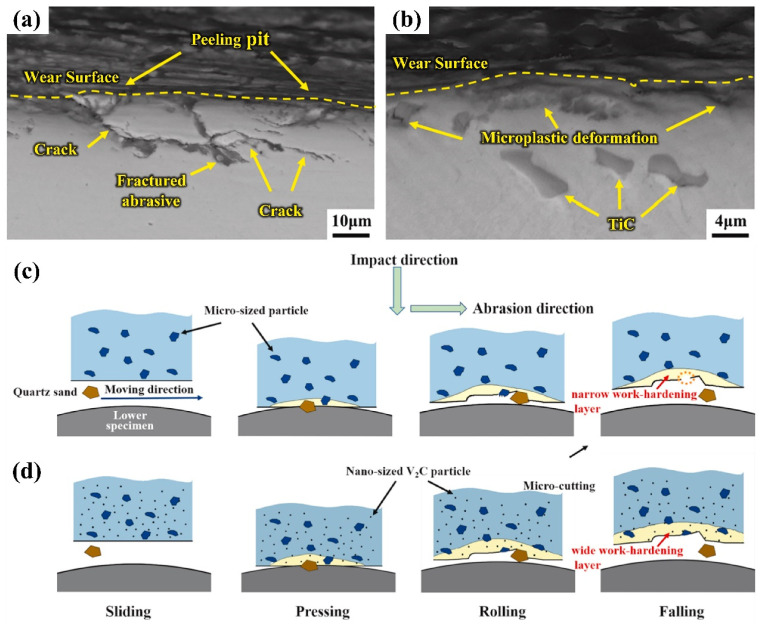
SEM micrographs of longitudinal sections of the worn surfaces of different steel types: (**a**) commercial wear-resistant steel (Hardox450); (**b**) bainitic wear-resistant steel (BS) [113]. Schematic diagram of the impact abrasive wear mechanism of high-manganese steel under different treatments: (**c**) without tempering treatment; (**d**) after tempering treatment [114].

**Figure 11 materials-17-03455-f011:**
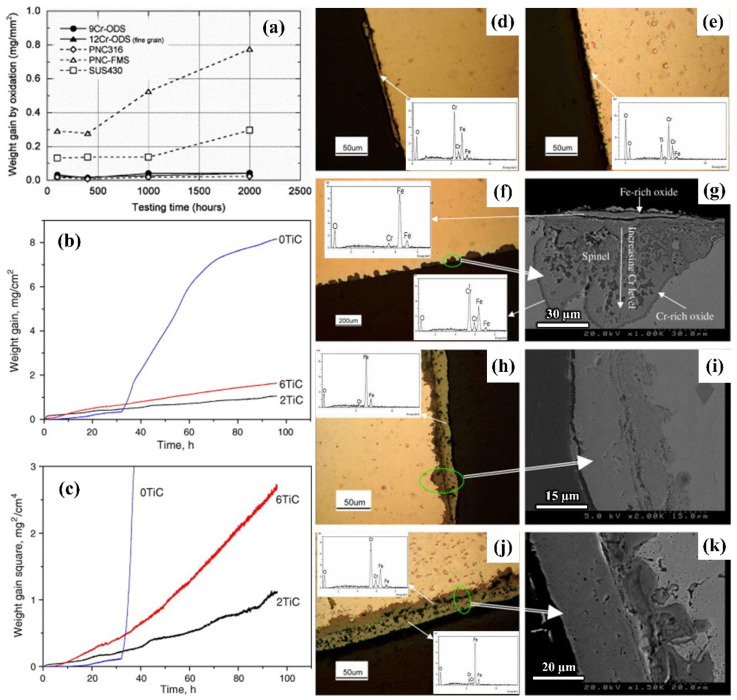
(**a**) Images of weight changes over time of different test steels at 1023 K [118]. Weight changes in 304SS steel with different TiC contents: (**b**) weight change curve with time (**c**) parabolic graph of weight change with time. Oxidation cross-sectional images of different TiC contents at different times: (**d**) 304SS, 48 h; (**e**) 304SS–6TiC, 48 h; (**f**,**g**) 304SS, 96 h; (**h**,**i**) 304SS–2TiC, 96 h; (**j**,**k**) 304SS–6TiC, 96 h [119].

## Data Availability

Not applicable.

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
