# Peer review of "The Main Failure Modes of Hot-Work Die Steel and the Development Status of Traditional Strengthening Methods and Nano-Strengthening Technology"

_materials, 2024, doi:10.3390/ma17143455_

Round 1

Reviewer 1 Report

Comments and Suggestions for Authors

The authors submitted the manuscript “The main failure modes of hot work die steel and the development status of traditional strengthening methods and nanostrengthening technology” to the journal “Materials”. The work is very valuable for researchers who work with hot work die steels. The manuscript can be published but requires major revision. All my suggestions are presented below.

1. English can be improved e.g. “Hot-work die steels are steel alloys..”

Steels are steels. Please modify it.

2. Line 40. Lack of distance.

3. Line 58. Lack of distance. “deformation[4]”

4. Figure 1. The scale bars on images c, d, e are not visible very well. It can be improved.

5. Figure 1. The SAED patterns are low quality. I do not see an indexing of patterns and information about the zone axis.

6. Figure 1. Lack of the scale bar on image f.

7. Lines 108-111. “Figure 1(f) shows the SEM images of thermal fatigue cracks in the test steel after 20,000  cycles of immersion testing and Figure 1(f 1 -f 5 ) shows EDS images of chemical elements around thermal fatigue cracks. As can be seen, there were higher Al and Si concentrations at the crack tip, indicating that the crack tip was filled with molten aluminum alloy. Additionally, the traces of C and O were also found at the crack tip.”

In Figure 1 I do not see C distribution.

8. Line 142. “EDS images of different chemical elements around thermal fatigue cracks: (f 1 ) Fe; (f 2 ) O; (f 3 ) Si; (f”

I recommend changing it to “distribution maps of selected alloying elements”

9. Figure 2c. The quality and size of the font need to be improved.

10. SEM-EDX spectra in Figure 2 are very low quality. I do not see anything on the images. I recommend modifying this part.

11. Figure 2. I recommend adding scale bars manually using ImageJ software, especially in Figures 2a, b, 2j and 2k.  

12. Figure 3. The size of a), b), c) symbols are in different sizes. I recommend to uniform it.

13. Line 267. “BSE-EDS line scan images of different steel types:”

Please modify this part because using BSE-EDS is a wrong nomenclature. In the open literature, the widely used version is: SEM-EDX, SEM-BSE, SEM-SE.

14. Line 270. “..ESD images..”.

I do not know what is ESD image. Please use the scientific nomenclature.  I see many mistakes like this. Please check the manuscript more carefully.

15. Figures 4a and 4b represent the stability of the phase with increasing temperature under equilibrium conditions.

16. Figure 6. “XRD analysis images”

I recommend using X-ray diffractograms or XRD spectra.       

17. Line 561. “SEM image of nanopowder”

I recommend changing it to nanopowder morphology.

18.Line 586. “without nano-TiC parti cles added;”

Please check again the style.

19. Line 590. “with nanoceramic additives”

Do you mean the nanoparticles?

20. Lines 606-607. “with the addition of 0 wt.% TiC and 0.1 wt.% TiC after casting”

Addition of 0 wt.%? Please check again the manuscript.

21. Lines 676-677. Do you mean curve or curves?  

22. Line 709. “TiB2” It should be corrected into TiB2.

23. Line 735. “nano-scale V2C precipitation”

I recommend changing it to V2C carbides.

24. Line 751. Please improve your style. Double space.

25. Figure 10. In the text you used V2C precipitation but in Figure 10d is V2C particles. Please uniform the style in the whole manuscript.

26. Figure 11. The SEM-EDX spectra are very low quality.

27. Figure 11. Scale bars in SEM-SE images are not visible. I recommend making scalebars manually with ImageJ software.

28. Reference [1] needs to be changed in style like the next positions.

Comments on the Quality of English Language

The manuscript should be improved before publication. 
I indicated the main grammar mistakes in my comments to the authors. 

Reviewer 2 Report

Comments and Suggestions for Authors

The manuscript titled "The main failure modes of hot work die steel and the develop-2 ment status of traditional strengthening methods and nano-3 strengthening technology" may be published taking into account the following comments:

1. In most cases, no spaces are used between the text and the literature reference. For example: "...dies[1]...". This should be corrected throughout the text.

2. H13 steel must be marked at least once in the text according to EN ISO.

3. When analyzing the oxidation process, refer to the Pilling-Bedworth relationship. There are many scientific papers describing this relationship. (DOI: 10.1088/2053-1591/ab5ea9, DOI: 10.1016/j.commatsci.2024.112925).

4. Figure 1 (c1, c2, c3, d1, d2, d3, e1, e2, e3) - illegible scale. This needs to be improved.

5. Figure 1 (d3, e3) – diffractions and descriptions illegible. This needs to be improved.

6. Figure 1 (f) – not to scale. This needs to be improved.

7. Figure 2 - not legible at all (scale, descriptions). This needs to be improved.

8. Figure 3. The scale for Al, Fe, O, Cr is illegible. This needs to be improved.

9. Figure 9. The signature should be subscript "...TiB2...".

10. Figure 9. There is no space between the parameter and the unit in the axis description.

11. Figure 11. (g, i, k) illegible scale. This needs to be improved.

12. Figure 11. (d, e, f, h, j) analyzes difficult to read. This needs to be improved.

Reviewer 3 Report

Comments and Suggestions for Authors

This review analyzes the failure modes of hot work die steel and outlines four traditional strengthening methods: optimizing alloying elements, electroslag remelting, increasing the forging ratio, and enhancing heat treatment processes. Additionally, a new nano-strengthening method, involving the addition of nanoparticles to molten steel for uniform dispersion, is introduced to improve the performance and longevity of these steels.

This review is very well done. Like any review, its purpose is to verify if all points are covered and balanced. This review achieves that. The structure of this review is relevant, and the list of references is quite comprehensive. The figures are of high quality and effectively complement the text."

However

In this review, the main failure modes of hot work die steel were analyzed. However, surface topography aspects enabling damage measurement were not addressed. This review aims to enhance the understanding and application of surface aspects to improve the service performance and lifespan of hot work die steel.". Please add it.

In the bibliography, several references are cited in the format [5–8]. It is important to justify each reference individually, This communicates the need to provide detailed justification or explanation for each cited reference in the bibliography.

Figure 10 b, Error, Plastic deformation, not plastite

Round 2

Reviewer 1 Report

Comments and Suggestions for Authors

The authors submitted the manuscript “The main failure modes of hot work die steel and the development status of traditional strengthening methods and nanostrengthening technology” to the journal “Materials”. The work is very valuable for researchers who work with hot work die steels. The authors included all my suggestions in the revised version. This manuscript can be published after minor revision. My last fine suggestions are:

1. On Figure 3 you write “SEM-BSE line  scan  images  of  different  steel  types”. Please take into account that the SEM-BSE is imaging of the microstructure due to Z contrast. The SEM-EDX corresponds to the chemical composition analysis via energy-dispersive X-ray spectroscopy. You should not mix this because it should be clearly written what you want to show.

2. Line 88. Please indicate more precisely the nomenclature. The steel AISI H13 is not ENISO: 40CrMoV5, as you write. It is a tool steel and is characterized by the standard ISO 4957. Moreover, you need to reference the standard ISO.

3. Line 277. The sentence “The surface topography plays an important role in materials engineering, in particular in the case of oxidized surfaces” is trivial and imprecise. Everything is important in materials engineering. I recommend rewriting this.

Reviewer 2 Report

Comments and Suggestions for Authors

The article has been corrected according to suggestions. I have no further comments.

Author Response

Thank you for taking the time to review my manuscript amidst your busy schedule. We have carefully revised it and hope everything goes smoothly!